# Preparation of Thermoplastic Polyurethane (TPU) Perforated Membrane via CO_2_ Foaming and Its Particle Separation Performance

**DOI:** 10.3390/polym11050847

**Published:** 2019-05-10

**Authors:** Chengbiao Ge, Wentao Zhai, Chul B. Park

**Affiliations:** 1School of Materials Science and Engineering, Sun Yat-sen University, Guangzhou 510275, China; gechengbiao08@126.com; 2Ningbo Key Lab of Polymer Materials, Ningbo Institute of Materials Technology and Engineering, Chinese Academy of Sciences, Ningbo 315201, China; 3Microcellular Plastics Manufacturing Laboratory, Department of Mechanical and Industrial Engineering, University of Toronto, Toronto, ON M5S 3G8, Canada

**Keywords:** thermoplastic polyurethane, foaming, perforated structure, membrane, filtration

## Abstract

The way in which a perforated structure is formed has attracted much interest in the porous membrane research community. This novel structure gives materials an excellent antifouling property as well as a low operating pressure and other benefits. Unfortunately, the current membrane fabrication methods usually involve multi-step processes and the use of organic solvents or additives. Our study is the first to offer a way to prepare perforated membrane by using a physical foaming technique with CO_2_ as the blowing agent. We selected thermoplastic polyurethane (TPU) as the base material because it is a biocompatible elastomer with excellent tensility, high abrasion resistance, and good elastic resilience. Various processing parameters, which included the saturation pressure, the foaming temperature, and the membrane thickness, were applied to adjust the TPU membrane’s perforated morphology. We proposed a possible formation mechanism of the perforated membrane. The as-prepared TPU membrane had good mechanical properties with a tensile strength of about 5 MPa and an elongation at break above 100%. Such mechanical properties make this novel membrane usable as a self-standing filter device. In addition, its straight-through channel structure can separate particles and meet different separation requirements.

## 1. Introduction

A porous structure within a polymeric membrane plays significant roles in such filtration areas as cell separation [1], water purification [2,3], gas filtration [4], battery separation [5], cell harvesting [6], and filtrate collection [7]. Up until the present, many techniques have been developed to prepare membrane materials, among which fibrous membrane and perforated membrane have been the most commonly produced products [8]. Fibrous membrane has been deemed to be an effective medium for filtration given its tunable fiber diameter, high porosity, remarkable specific surface area, and interconnected porous structure [9,10]. Nevertheless, membrane fouling greatly challenges its applications [3,9]. This is due to its sponge-like or tortuous pore channel structure, which makes it difficult to clean within the material’s structure [11]. 

Multiple scientific and engineering methods have been used to enhance the membrane’s antifouling property in the application process. Among these investigations, it was found that a perforated structure gave the material an excellent antifouling property [12]. These as-prepared membranes have recently been used in the filtration of both lager beer [13] and of bacteria [14]. This has been due to their unique characteristics; that is, a straight pore channel and a low operating pressure [11,15]. The following techniques have been used to prepare perforated membranes: the breath figure method, anodization, lithographic microfabrication, colloidal crystal assembly, emulsion templating, and microphase separation of block copolymers [1,16]. However, an organic solvent or an additive has been most commonly used in the fabrication of the noted materials. Typically, to obtain a perforated membrane using the breath figure method, the polymer needs to be dissolved in carbon disulfide. Next, it is placed on the water’s surface, and then it is carefully transferred to the support device [1]. The environmentally hostile nature of solvents and/or additives like carbon bisulfide, chloroform, and tetrahydrofuran limits their large-scale preparation. In addition, their harsh operating conditions present further drawbacks [17]. Also, some of the as-prepared perforated membranes’ mechanical properties are insufficient for them to be used as self-standing filtration media, which require support from high-strength materials during the separation process [15,18]. Thus, the search for a simple, efficient, and feasible way to prepare perforated membrane for use in the filtration field is very meaningful.

It’s well known that the physical foaming process is a high-efficiency technique to prepare porous materials, and it has enormous advantages. This is because it is both environmentally-friendly, cost-effective, and its features are straightforward [19,20]. Carbon dioxide, nitrogen gas, water vapor, and other physical foaming agents have been widely adopted for use in the preparation of polymeric foams of polypropylene, polycarbonate, polyethylene terephthalate, and polylactic acid [21,22,23,24] in various foaming processes such as batch foaming, foam extrusion, and foam injection moulding. The interconnected open-cell structure can be formed via the previously noted techniques under high temperature and pressure conditions [25,26]. However, since CO_2_ has a high diffusion rate from the polymer surface, the formation of a dense skin layer is a very common phenomenon [19,27,28]. Thus, the structure; that is, the interconnected open-cell structure and the dense skin layer show that a perforated membrane cannot be created using the existing foaming technology. The nucleation of cells mainly occurs in the first few seconds of the foaming process, involved in two mechanisms of homogeneous nucleation and heterogeneous nucleation; for the former mechanism, it is a spontaneous and non-filler assisted formation process without cell precursor; for the latter nucleation process, it works normally with the help of the filler interface, shear, ultrasound, and the crystalline regions, etc. [29,30,31,32]. In our study, which makes use of the heterogeneous nucleation effect on the polymeric material’s surface, we have presented a novel way to prepare perforated membrane.

As previously noted, a membrane with poor deformation ability cannot be applied to the filtration process in a self-standing way. Thermoplastic polyurethane (TPU), a type of elastomer, shows potential for use in the preparation of perforated membrane because of its excellent tensility, good elastic resilience, and excellent biocompatibility [33,34,35]. Our study describes, for the first time, a simple fabrication method for perforated membrane, which is based on the heterogeneous nucleation effect on the TPU’s surface. By sandwiching the TPU film between two polyimide (PI) films blown with CO_2_, this structure provides heterogeneous nucleation sites at the interfaces between the TPU film and the PI film. At the same time, a perforated membrane with different morphologies can be obtained by adjusting the foaming condition. Based on our differential scanning calorimetry results, we have proposed a formation mechanism for perforated membrane. In the last part of our paper, we discuss the influences of cell structures on the filtration function with separation of the polystyrene microsphere. 

## 2. Experimental

### 2.1. Materials

The polyester TPU (380A) was obtained from Austin, Co., Zhangjiagang, China in a pellet form. Within TPU matrix, the hard segment is composed of 4,4’-diphenylmethane diisocyanate with 1,4-butylene-glycol as the chain extender, and the soft segment is poly(1,4-butylene adipate). The polyimide (PI) film was received from Kapton with a thickness of about 45 μm, which was derived from the condensation polymerization of the pyromellitic dianhydride (PMDA) and the 4,4’-diaminodiphenyl ether (DDE). The CO_2_ physical blowing agent was obtained from Ningbo Wanli Gas, Co., Ningbo, China with a 99.9% purity. The polystyrene microsphere used for the filtration evaluation was prepared using a method obtained from the literature [36]. 

### 2.2. Preparation of the TPU Sandwich Structure Film

To fabricate the PI/TPU/PI sandwich structure film, the as-received TPU pellets were dried in a vacuum oven at 60 °C for 5 h to remove the moisture, and were then compression molded between two PI films. Typically, 3 g TPU pellets were dispersed onto the PI film, which measured 20 cm × 20 cm. Further, a PI film of the same size was put on the above the TPU pellets. Next, the films were loaded into a hydraulic press, and were heated at 190 °C for 3 min. The PI/TPU/PI film then formed under 15 MPa after 3 min. After this process, a TPU layer with a thickness of about 20 μm can be obtained. Notably, by changing the pressure during the hot compression step, different TPU layer thicknesses can be achieved.

### 2.3. Preparation of the TPU Perforated Membrane

The PI/TPU/PI film was enclosed in a stainless steel chamber flushing with low-pressure CO_2_ for 2 min. Then, the film was placed under the designated pressure and room temperature for 12 h to ensure the equilibrium adsorption of the CO_2_. Once the saturation process was finished, the chamber was depressurized at a rate of 0.5 MPa·s-1. Then, within 30 s, the film was transferred to a hot bath with polydimethylsiloxane (PDMS) as the heating medium at a fixed temperature for 10 s to foam. The foamed membrane was put into an ice bath to fix its cell structure. The residual PDMS on foams’ surface was removed by cleaning.

### 2.4. Characterizations

We used a scanning electron microscope (SEM; Zeiss EVO18, Aalen, Germany) to observe the morphologies of the membranes and the polystyrene microsphere. For the membrane specimens, they were prepared by using a sharp-edged knife, and were then sputter-coated with a thin platinum layer. To see the polystyrene microsphere’s morphology, the microsphere’s suspension was dropped onto a silicon wafer. Then the dispersion medium, which was ethanol, evaporated in the atmospheric environment. The accelerating voltage of the scanning electron microscope was 20 kV. We determined the diameter and size distribution of both the cells and the microspheres by analyzing the SEM images.

The thermal behavior of the TPU thin films was recorded before and after the saturation process, using differential scanning calorimeter measurements. The melting temperature (*T*_m_) and the enthalpy of fusion (∆*H*_m_) were measured via the Diamond DSC (PerkinElmer, Waltham, MA, USA), and had been obtained from the first heating scan. That scan ranged from 25 to 220 °C at a heating rate of 10 °C·min^−1^ in a nitrogen environment. Before the measurement was taken, the saturated TPU thin film was allowed to degas at room temperature for 48 h.

We used a universal testing machine (Instron 5567, Boston, MA, USA) for the tensile test. The specimens were under a strain rate of 200 mm·min^−1^. They measured 30 mm in length, 10 mm in width, and were about 0.02 mm thickness. The test was done at the room temperature. Each specimen’s tensile strength was acquired based on five values.

The peel strength between the PI film and the TPU film was obtained by using a universal testing machine (Instron 5567) according to ASTM D1876 and the reference [37]. The distance between two clamps was 50 mm, and the peel rate was 100 mm·min^−1^.

## 3. Results and Discussion

### 3.1. Preparation of the TPU Perforated Membrane

Unlike the breath figure method, anodization, lithographic microfabrication, colloidal crystal assembly, emulsion templating, and the microphase separation of block copolymers, a typical feature of the cellular materials fabricated by physical foaming technique is this: the as-received samples have closed cells that are insulated from the surrounding cells or the interconnected open cells [25,38,39,40]. Therefore, to obtain the perforated membrane, the traditional foaming process needs to be improved. Traditionally, a polymer matrix blended with fillers or used crystalline domains has been used in the foaming process. Also, the foaming ability of the as-prepared foamed material is far beyond that of the neat polymer because these domains act as heterogeneous nucleation sites that reduce the energy barrier for cell nucleation [24,29,38]. As reported by Kakroodi [41], only a 3 wt % of microfibrils dramatically improved the foaming ability of poly(lactic acid). By applying this nucleation mechanism, we designed, for the first time, a technique that causes a heterogeneous nucleation effect on the material’s surface. This can then be used to successfully prepare a perforated membrane.

To improve the mechanical properties of the perforated membrane, we selected TPU, a thermoplastic elastomer, as the base material in our work. Thermoplastic polyurethane is a copolymer consisting of both hard and soft segments [42,43]. However, due to their polarity differences, the polar hard segments aggregate into hard domains and form crystalline areas [44,45]. As previously noted, these areas act as nucleation sites within the polymer matrix to effectively generate small cells. Such cells in the cross-section do not help to form a straight and cylindrical perforated structure. In Figure 1, the DSC thermographs of the as-received TPU specimens before and after the saturation process under 2.0 MPa are shown in association with a melting peak of about 190 °C. This melting peak is related to the melt that occurs in the hard segment micro-crystalline area [46]. Before the saturation process, the *T*_m_ and the ∆*H*_m_ were, respectively, 188.9 °C and 1.24 J/g. After the saturation process, the *T*_m_ and the ∆*H*_m_ increased, as we had expected they would, and are shown in Figure 1. They were 190.6 °C and 1.79 J/g, respectively. This result showed how CO_2_ could facilitate the chains’ movement; thus, improving the hard segment domain’s perfection [28]. At the same time, this phenomenon illustrated how the crystalline area always exists in the TPU’s matrix, either before or after the saturation process. Even with the heterogeneous nucleation’s influence, the cell sizes within the foamed material obtained by the conventional technique are usually on a micron scale [24,47]. Thus, we envisaged reducing the material’s thickness to approximate the cell sizes needed to produce a single-layer cell structure; that is, a perforated structure.

Figure 2 shows the preparation process. Typically, the base material, meaning the TPU pellets, was dispersed between two PI films and compressed under 15 MPa and 190 °C to achieve a thickness of about 20 μm TPU film (Figure 2a,b). This was close to the cell size of the foamed materials obtained through the traditional technique [24]. The sandwich film was saturated in a high-pressure chamber at 25 °C for 12 h (Figure 2c), allowing sufficient CO_2_ dissolution in the TPU’s matrix. The saturated TPU was removed from the chamber (Figure 2d) and placed into a high-temperature oil bath to foam via the temperature-induced foaming method (Figure 2e). The presence of large amounts of interface area between the TPU and the PI films could act as the heterogeneous nucleation sites. These tended to enhance cell nucleation on the TPU’s surface, which resulted from the lower energy barrier required for heterogeneous nucleation. Consequently, a perforated membrane could be easily prepared (Figure 2f) during the foaming.

Figure 3 shows a typical TPU perforated membrane. Using the interface between the PI and TPU films to generate the heterogeneous nucleation sites, the straight and cylindrical pore channel membrane was successfully prepared by this foaming process. In Figure 3a, the highly ordered straight pore channel was observed in the cross-section, and was characterized by the large inner diameter and the small TPU surface opening size. This membrane’s thickness was about 20 μm. Morphological observations of the TPU’s surface (Figure 3b) further confirmed the perforated structure, which was highly suitable for the particle separation. The cells located on the surface obviously controlled the accuracy of the separation. Figure 3c shows the cell diameter and its distribution on the surface. This was in the range of 6.4–17.6 μm, and the average value was 11.6 μm.

### 3.2. Influence of the Saturation Pressure and Foaming Temperature on the TPU Perforated Membrane’s Morphology

The above-noted method of obtaining the perforated membrane was successful under a 2.0 MPa saturation pressure. However, it is well known that a wide processing window is highly important in industrial applications and in scientific research. We used this novel method for the first time to examine the effects of the saturation pressure and foaming temperature on the traditional foaming process. Further, we investigated the influence of these factors on the perforated membrane’s morphology. Figure 4 shows the morphologies of the cross-section and the surfaces of the TPU’s perforated membrane saturated under different pressures. Under 2.5 and 3.0 MPa saturation pressures, the highly ordered straight pore channel could still be formed in the cross-section. Under the 2.5 and 3.0 MPa saturation pressures, and even including the morphology shown in Figure 3, the cell size on the material’s surface was significantly lower than the diameter of the straight pore channel in the cross-section. In the foaming process, the sizes of the cells that formed using the heterogeneous nucleation mechanism were much lower than those that formed during the homogeneous nucleation process. This is due to the heterogeneous nucleation mechanism’s low energy barrier [38,48,49,50]. Apparently, the material’s surface morphology was caused by the dominant heterogeneous nucleation mechanism. Although the TPU used in our study formed a crystalline area, its melting enthalpy was very low. Consequently, the large diameter straight-through channel in the cross-section indicated that the homogeneous nucleation scheme was dominant in this region. We have discussed this in detail later in this paper. In addition, under the 2.5 and 3.0 MPa saturation pressures, the cell size ranges and the average cell size of the materials’ surfaces were 2.7–10.5 μm and 6.4 μm, and 1.2–8.5 and 5.1 μm, respectively. With reference to the sample morphology saturated under 2.0 MPa, the cell size decreased gradually in conjunction with the increased saturation pressure. The reason was that an increased saturation pressure tended to increase the CO_2_ solubility within polymer matrix, which reduced the barrier of cell nucleation, and therefore resulting in the increased cell density and the decreased cell size [51,52].

The foaming temperature is clearly a very important parameter, with respect to the appearance of the foamed material’s morphology in the foaming process. Usually, the foaming condition, along with the increased temperature, greatly improves the foaming ability [24,53], but a too high foaming temperature will cause the collapse of foamed material as well as other structural damage [33]. As previously noted, low saturation pressures produced large cells, which could then generate a very thick perforated membrane. Figure 5 shows the morphology of the cross-section and the corresponding surface after the foaming process. The saturation pressure was 1.5 MPa. In Figure 5(a,a1), the cells cannot be seen either on the surface or in the cross-section. However, when the foaming temperature was increased to 120 °C, the surface and the cross-section presented some cells, but the perforated structure still had not formed. A clear perforated structure did not appear until the foaming temperature had risen to 130 °C. However, at a high temperature of 140 °C the cell structure was significantly damaged, as proven by the cell collapse (Figure 5d). In Figure 5c the perforated membrane was about 40 μm thickness.

Figure 6 shows the statistical analysis of the surface morphologies of the foamed materials obtained at different foaming temperatures. It is obvious that as the temperature increased from 120 to 130 °C, the average size and the maximum size of the cells increased, and, then from 130 to 140 °C, their sizes decreased. For example, the average cell size increased from 17.5 to 20.3 μm, and then it decreased to 15.1 μm. Meanwhile, the maximum cell size increased from 23.8 to 39.7 μm, and then it decreased to 27.0 μm. This type of tendency also helped to explain why a perforated structure could be formed at a 130 °C foaming temperature. At 130 °C large cells form, and these can cross the 40 μm thickness membrane. The increased cell sizes, which followed the increased foaming temperature, occurred mainly because the high foaming temperature reduced the cell wall’s strength, and then facilitated cell coalescence [38,54]. The morphology at the 140 °C foaming temperature was related to the cell structure’s collapse. 

### 3.3. Formation Mechanism of the Perforated Membrane

The above results indicated that the formation of a perforated structure benefits from cells that can move across the cross-section. Figure 7 shows a cross-section of the membranes with different thicknesses formed under different saturation pressures and foaming temperatures, and further verifies the cell’s influence on the cross-section of the perforated structure. As shown in Figure 7a, an undesirable double-row cell structure was formed even with a low membrane thickness of 21 μm, when a high saturation pressure of 3.5 MPa was used. As the saturation pressure was further increased to 4.0 MPa, a conventional closed-cell structure was formed inside the membrane because too many cells generated. These results strongly indicate that a saturation pressure of below 3.0 MPa would be necessary to form a desirable single-row cell structure to make a perforated membrane when the membrane thickness is approximate 20 μm. When a higher saturation pressure was used, a higher cell density was induced [55] and, consequently, small cells in the cross-section that have formed under a high saturation pressure were unable to effectively cross a membrane of 20 μm thickness. 

Figure 7d–g show the effects of the perforated membrane’s thickness under different foaming conditions. As noted above, a low saturation pressure can form large cells because of a low cell density, which can cross even thick membranes. A 70 μm thickness of perforated membrane cannot effectively form under a saturation pressure of 1.5 MPa. The thick skin layer prevents the cells in the cross-section from coalescing with those on the surface. Similarly, after the saturation pressures were increased to 2.0, 2.5, and 3.0 MPa, a double-row of cell membranes with thicknesses of about 39, 34 and 28 μm was, respectively, obtained. The cells in the cross-section effectively gathered with those on the surface, but the cell coalescence in the cross-section seemed to be negligible due to the thick cell wall. This further showed that cells which can move across the cross-section are needed to form a perforated structure.

However, when the pressure increased above 3.5 MPa, the cells on the surface did not effectively gather with those in the cross-section. Many small cells on the surface just presented a half-cell structure, that is, half of a normal cell (as pointed by the red arrow in Figure 7c). Figure 7h,i show that, under higher saturation pressure, i.e., 4.5 MPa, the surface forms numerous half-cells and a concave structure without coalescence with those in the cross-section. This indicates that the cells on the surface may have played a role for the formation of the perforated structure. When a low pressure was adopted, the large cells can be obtained, which can coalesce with those in the cross-section.

The cell formation process includes nucleation, cell growth, and cell coalescence [56]. We believe that the formation of a perforated structure, possessing the straight pore channel with a large diameter in the cross-section and small openings on both upper and lower surfaces, may be due to the difference in the cell-nucleation mechanisms that occur between the surface and the cross-section of the TPU film, and the shear force and the friction between the TPU film and PI film. In our study, the TPU film’s upper and the lower surfaces adhered to the PI film. As shown in Figure 8, the peel strength was about 29.6 N, which demonstrates that a well-defined interface bonding was formed between TPU and PI films, and then provided a lot of heterogeneous nucleation sites to form small cells. Liao et al. [57,58] investigated the foaming behavior of two-phase blend and reported that these cells on the surface or at the interface were due to the low energy barrier of heterogeneous nucleation, which facilitated the formation of small cells. On the other hand, since foaming occurred only in the TPU layer during the foaming process, this will cause a relative movement between the TPU layer and PI layer. Under the influence of the existence of adhesion, the foaming behavior will produce friction or shear force at the interfaces, which would also be beneficial for the formation of those cells on the surface of the TPU layer. Furthermore, the matrix of the TPU layer also provided cell-nucleation sites, most likely at the interfaces of the soft segments and the hard segments [29,59]. We note that the melting enthalpy in reported TPU specimens was about 40 J/g [29,59] while the melting value of the TPU resin we used was only about 1 J/g. Therefore, the cell nucleation mechanism within the TPU’s matrix played a leading role to form relatively large cells.

The polymer matrix with dissolved CO_2_ was foamed by a temperature-induced method. In this work, under the influence of the heterogeneous nucleation, the shear force, and the friction on the surface, a low saturation pressure was adopted to produce single-layer large cells in the cross-section and no small cells on the surface of membrane. The cell wall’s strength may not have been high enough to resist cell coalescence in a high foaming temperature environment [38]. It seemed that the cross-section cells were coalesced with those on the surface and, consequently, the perforated structure was formed. However, given the current experimental results, more experiments are needed to understand the formation mechanism of perforated structures in depth.

### 3.4. Particle Separation

Most of the perforated membranes cannot be used as self-standing filtration media due to their poor mechanical property. They must be supported by high-strength materials in the separation process [15,18]. In a TPU-perforated membrane, given the base material’s good mechanical properties, it should be possible to use this as a self-standing separation medium. Figure 9 shows the tensile properties of the as-prepared perforated membrane obtained under different conditions. These materials demonstrated good deformation behavior: the breaking elongation was above 100% strain and the tensile strength was about 5 MPa. They could be used as the self-standing separation material. It was also apparent that, with the surface cell size being decreased, the mechanical properties of the material, including the elongation at break and the tensile strength, were improved. They increased from 149% and 3.9 MPa to 191% and from 5.1 MPa to 222% and 5.5 MPa to 235% and 5.6 MPa, respectively. This result may have originated from the stress concentration around the large cells within the foamed material, which would induce failure initiation of the material [60].

For example, the TPU perforated membrane saturated under 3.0 MPa was used to separate the polystyrene microsphere so as to evaluate the filtration performance. Figure 10a shows a simple filter device. Based on the colour change in the suspension and filtrate; namely, the white suspension and transparent filtrate, this membrane can act as an effective separation instrument for water treatment and particle collection. The TPU-perforated membrane, without any support device, was placed directly between the syringe and the needle, and the operation process was implemented without external pressure. The suspension flowed through the TPU-perforated membrane, and the filtrate was collected in the glass bottle for the particle size statistics. In Figure 10b, the SEM images and the corresponding particle size statistics are shown both before and after the filtration. As we had expected, the decreased maximum size of the microsphere from 31.4 to 4.6 μm and the narrower size distribution were clearly seen. This result shows that the TPU perforated membrane can effectively remove large particles from the suspension.

In addition, the polystyrene microspheres can be filtered by using perforated membranes equipped with different surface morphologies to obtain filtrates that meet different requirements. Figure 11 shows the filtration effect of the perforated membranes, which were prepared under conditions where the saturation pressures were 1.5, 2.0, and 2.5 MPa, respectively. After filtration, the maximum size of the microspheres in the filtrate was about 29.0, 14.0, and 9.6 μm, and the size distribution had also decreased from what it was in Figure 11a–c. Obviously, the material had achieved a different filtration effect, and could meet different filtration requirements. Moreover, we believe that these TPU perforated membranes with straight and cylindrical structures may possess an excellent antifouling property relative to the fibrous membrane, which has a tortuous channel structure that makes it difficult to clean [11].

## 4. Conclusions

In our study, the straight through channel structure was successfully introduced into the TPU membrane by means of the heterogeneous nucleation’s effect on the material’s surface during the foaming process. The morphology of these perforated membranes could be easily regulated by changing the saturation pressure, the foaming temperature, and the membrane’s thickness. A high saturation pressure could generate a small channel membrane while a low value pressure could be used to prepare a thick membrane with large channels. The morphological analysis and the DSC results indicated that the membrane’s holes were formed mainly through the heterogeneous nucleation mechanism at the interfaces of the TPU and PI films. Although cell nucleation also occurred within the TPU film, the perforated membranes were successfully formed because of the small thickness of the membrane. In addition, the perforated membranes had good mechanical properties and could be used as self-standing separation devices. The particle separation results showed that the membranes could effectively filter microspheres and meet different requirements.

## Figures and Tables

**Figure 1 polymers-11-00847-f001:**
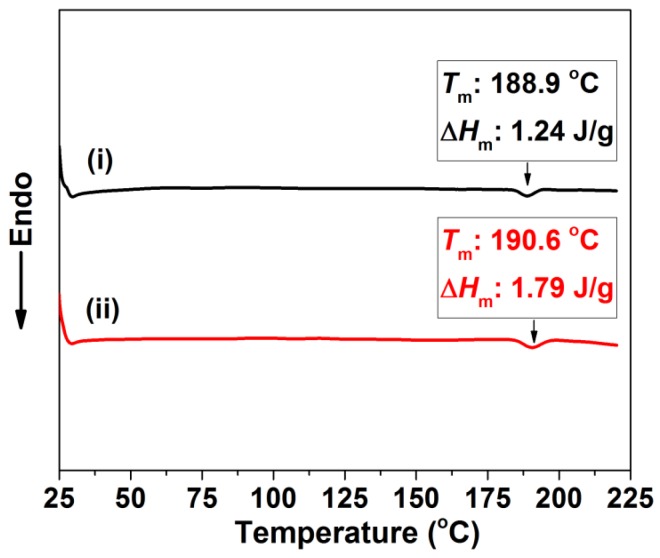
DSC thermogram of TPU specimens: (**i**) original TPU; (**ii**) TPU after saturating process.

**Figure 2 polymers-11-00847-f002:**
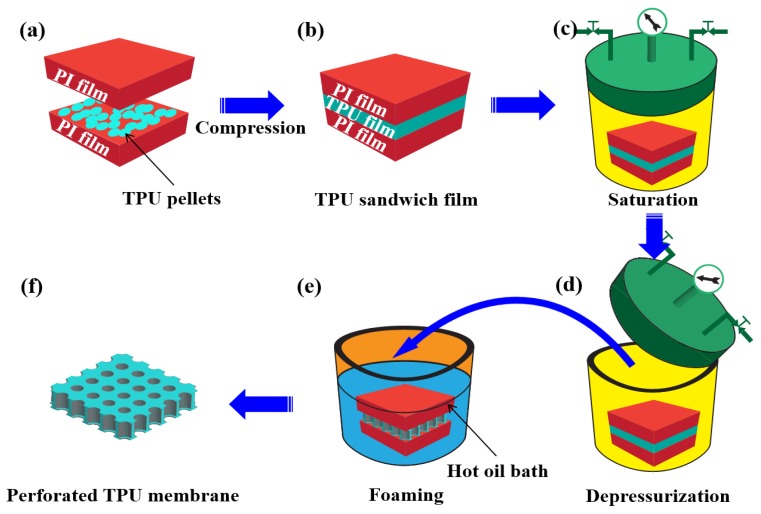
Schematic for preparation of TPU perforated membrane.

**Figure 3 polymers-11-00847-f003:**
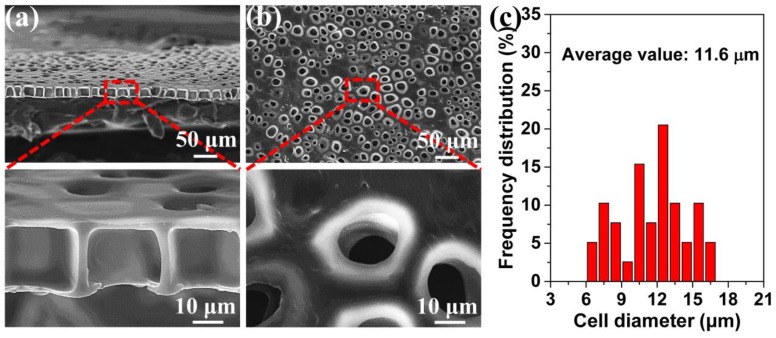
The TPU perforated membrane saturated under 2.0 MPa and foamed at 120 °C. (**a**) the cross-section, (**b**) the surface, (**c**) the average diameter of the cells on the surface.

**Figure 4 polymers-11-00847-f004:**
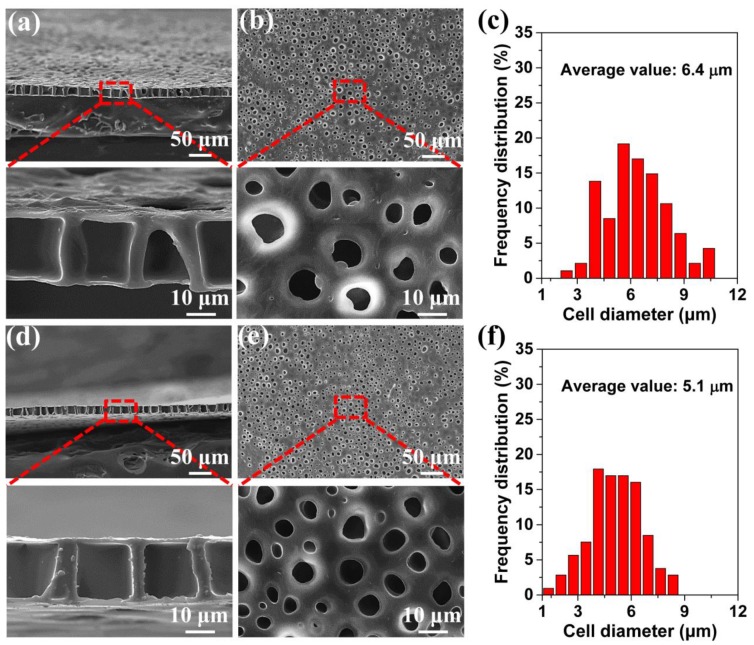
SEM images of the prepared TPU perforated membranes, the films were saturated under 2.5 MPa (**a**–**c**) and 3.0 MPa (**d**–**f**), respectively, and then were foamed at 120 °C for 10 s. (**a**,**d**) the cross-section, (**b**,**e**) the surface, and (**c**,**f**) the average diameter of the cells on the surface.

**Figure 5 polymers-11-00847-f005:**
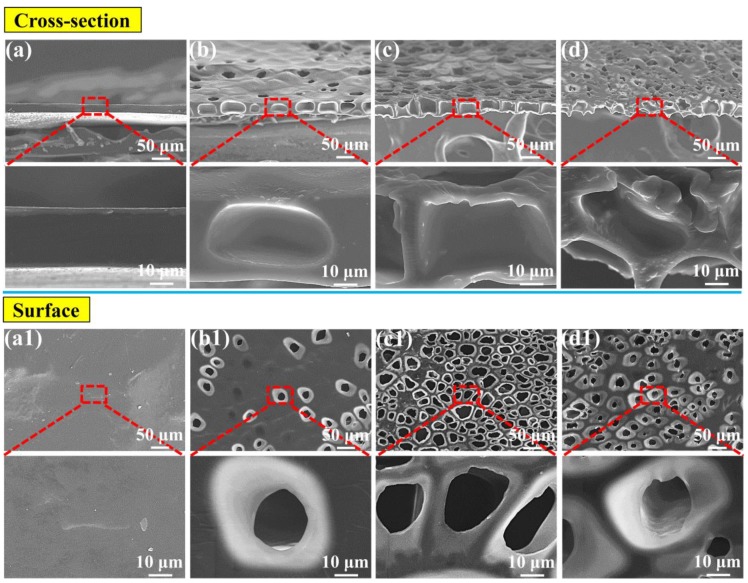
SEM images of the TPU perforated membrane saturated under 1.5 MPa: (**a**) the cross-section and (**a1**) the corresponding surface foamed at 110 °C. Following similar principle, (**b**,**b1**), (**c**,**c1**), as well as (**d**,**d1**) foamed at 120, 130 and 140 °C, respectively.

**Figure 6 polymers-11-00847-f006:**
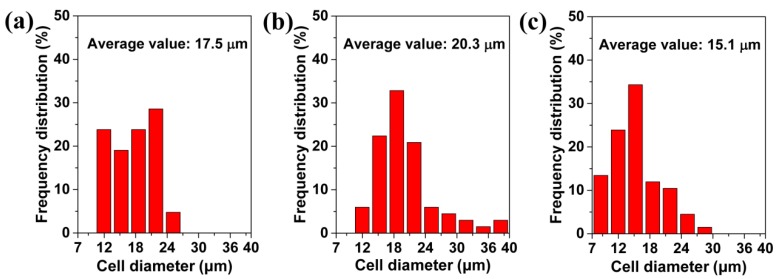
The average value of cells on the surface of TPU perforated membrane saturated under 1.5 MPa and foamed at different temperatures of: (**a**) 120 °C, (**b**) 130 °C, and (**c**) 140 °C.

**Figure 7 polymers-11-00847-f007:**
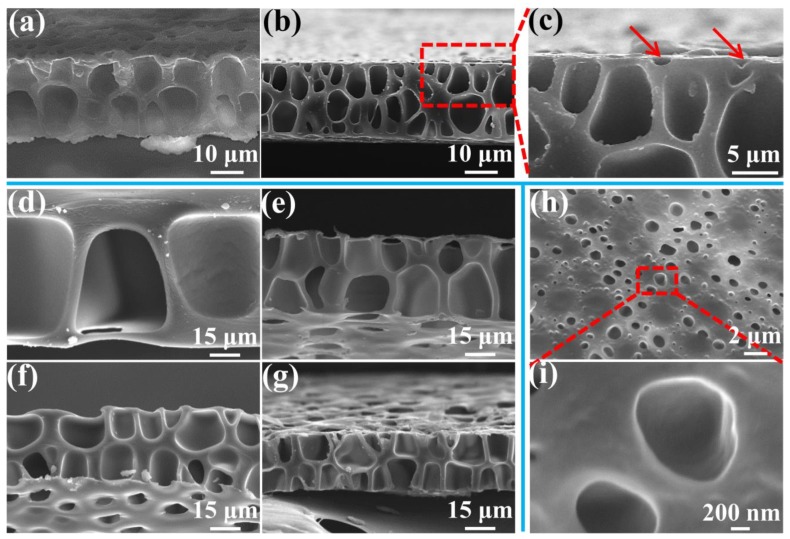
SEM images of the cross-section of the membrane with various thicknesses prepared under different saturation pressure and foaming temperature conditions: (**a**) 21 μm, 3.5 MPa, and 120 °C; (**b**–**c**) 23 μm, 4.0 MPa, and 120 °C; (**d**) 70 μm, 1.5 MPa, and 130 °C; (**e**) 39 μm, 2.0 MPa, and 120 °C; (**f**) 34 μm, 2.5 MPa, and 120 °C; and (**g**) 28 μm, 3.0 MPa, and 120 °C. SEM images of the surface of the membrane (**h**,**i**), obtained under 4.5 MPa saturation pressure and 120 °C foaming temperature.

**Figure 8 polymers-11-00847-f008:**
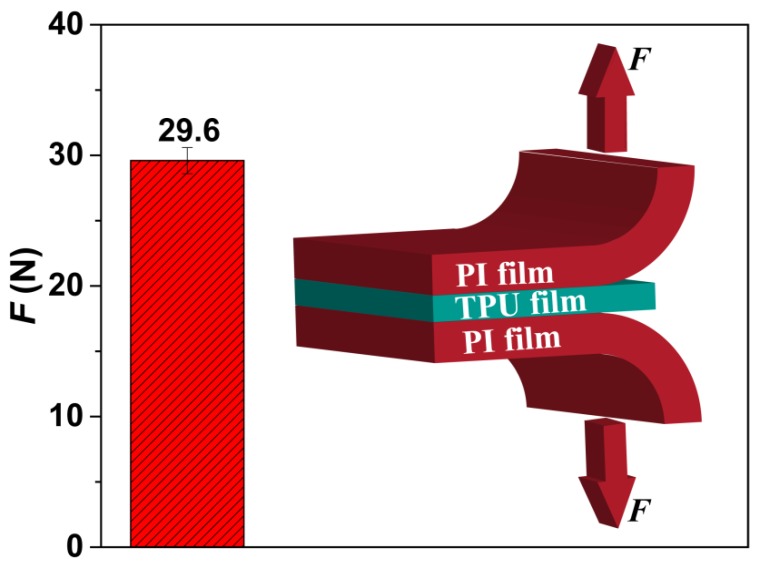
The peel strength of PI/TPU/PI sandwich film, including an insert schematic figure.

**Figure 9 polymers-11-00847-f009:**
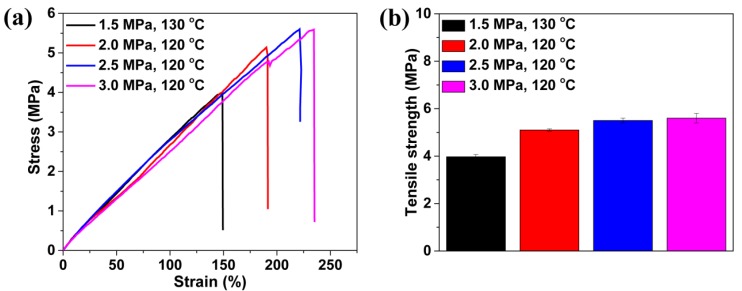
Tensile performance of TPU perforated membranes saturated and foamed under different pressures and temperatures. (**a**) typical stress-strain curves and (**b**) the tensile strength.

**Figure 10 polymers-11-00847-f010:**
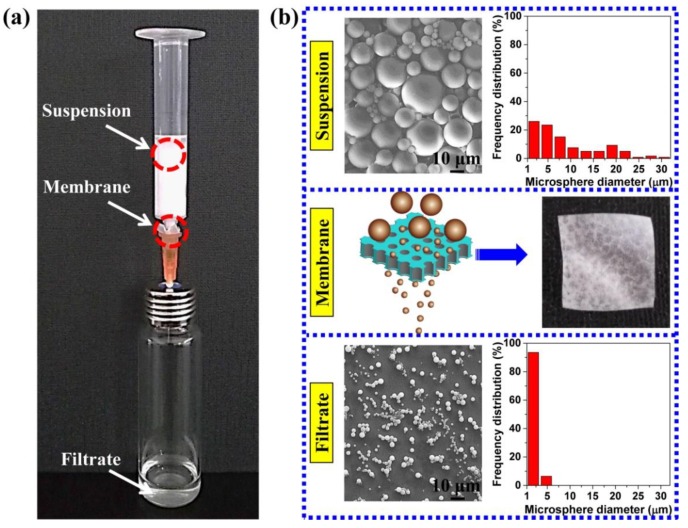
(**a**) the optical photograph of filtration process. (**b**) SEM images and optical photograph of suspension, filtrate and membrane.

**Figure 11 polymers-11-00847-f011:**
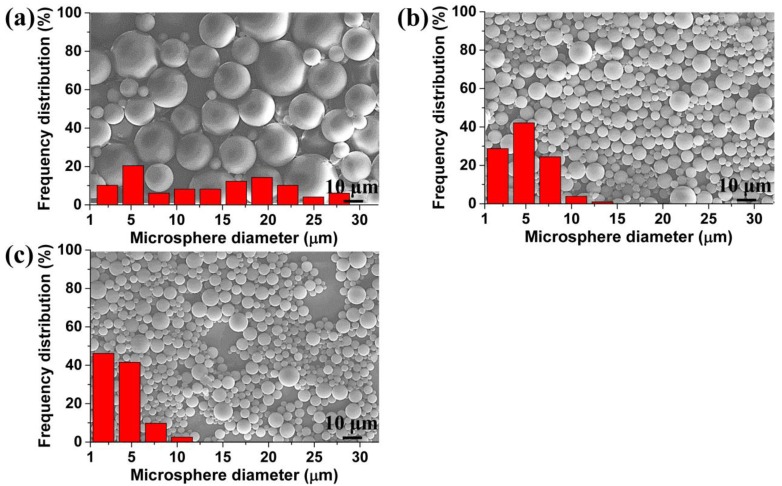
SEM images and size distribution of microspheres in the filtrate, obtained from the membrane with: (**a**) 20.3 μm, (**b**) 11.6 μm, and (**c**) 6.4 μm average cell diameter on the surface.

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
