# Peer review of "Preparation of Thermoplastic Polyurethane (TPU) Perforated Membrane via CO2 Foaming and Its Particle Separation Performance"

_polymers, 2019, doi:10.3390/polym11050847_

Round 1

Reviewer 1 Report

The manuscript entitled “Preparation of thermoplastic polyurethane (TPU) perforated membrane via CO2 foaming and its particle separation performance” describes the preparation of perforated membranes using CO2 as foaming agent. The methodology described is simpler than others described in the literature, however for its industrial application it would be necessary to go deeper into this methodology.

The present work is well presented and structured. Therefore, in my opinion, the present manuscript can be recommended for publication after the author take into account the following remarks and perform the experimental work to establish their conclusions.

In Figure 2, the authors present the DSC results of the commercial polyurethane used. The temperature range is 25ºC and 225ºC, to observe the fusion of the hard segment of the material. However, the authors should include the DSC of the material at lower temperatures to observe the fusion of the soft segment. In this way we could know, in a certain way, the chemical composition of the soft segment and its influence on the formation of perforated membranes.

The authors indicate that they introduce the polyurethane film into a hot PDMS bath to form the perforated membranes. However, the authors do not indicate if this process would lead to an adsorption of PDMS on the surface of the membrane. Is it necessary to remove siloxane residues?

Please, the authors should comment on this aspect.

Throughout the manuscript, the authors have been carried out a study of cell sizes and their distribution. However, the authors do not mention how they have conducted these studies. Authors should indicate the used software and how they established the conditions for their studies and their interpretation.

Author Response

Referee #1

In this paper, the authors have developed a novel membrane fabrication method using CO2 as a physical foaming agent. They have made a perforated membrane of a TPU which has the smaller pore diameter at the surface and the large diameter straight channel at the cross-section region. The paper explained that the interfacial adhesion, friction force and shear force between PI and TPU are responsible for the smaller surface pore size. The authors have also described how the different process parameter such as foaming temperature, saturation pressure, and the membrane thickness has an effect on the morphology and mechanical properties of the membrane. The paper is written well. I want to accept the paper with minor revisions.

Comment 1: In the abstract, the authors have mentioned that the perforated membrane enhances the antifouling properties of the membrane. Do you have any data that suggests that the TPU perforated membrane enhances the antifouling?

A1: As evidenced by reference 12, the perforated membrane materials have straight-through pore structure that allows the membrane material to well clean internal structure. For the fiber membrane and other materials, its interior has tortuous hole channel, resulting in the difficulty during the cleaning. From this point of view, the straight-through hole structure has good antifouling ability. A further research about the separation character of the prepared member will be carried out in near future.

Comment 2: In the material section, please provide the information about the type of TPU – the amount of the hard segment, amount of soft segment, type of polyol used for hard and soft segments, type of isocyanides used, and amount of hydrogen bonded polyurethane
A2: Sure. The TPU information has been given in the revised manuscript in line 93-95 of page 3.

Comment 3: Please provide the details about the homogeneous and the heterogeneous nucleation of CO2 in the polymeric matrix. Please add it in the introduction section. Please include how the enthalpy and entropic energy related to the activation energy of nucleation. Also, add how the activation energy reduces with the heterogeneous nucleation.

A3: Sure. The revised descriptions have shown in lines 71-78 of page 2.

Comment 4: Please provide the reasons or some hypothesis about why does the higher amount CO2 mean higher nucleation sites. Please add it in section 3.2.

A4: Sure. The revised descriptions have been indicated in lines 232-234 of page 8.

Comment 5: The line 307 says “ the melting enthalpy in these materials was about 40 J/g”, which material has melting enthalpy 40 J/gm PI or TPU? How it was measured?

A5: PI is an amorphous polymer, while TPU can be crystallized and its melting enthalpy of 100% crystals is 40J/g based on the references 29 and 59. The revised descriptions have been shown in lines 324-325 of page 11.

Reviewer 2 Report

In this paper, the authors have developed a novel membrane fabrication method using CO2 as a physical foaming agent. They have made a perforated membrane of a TPU which has the smaller pore diameter at the surface and the large diameter straight channel at the cross-section region. The paper explained that the interfacial adhesion, friction force and shear force between PI and TPU are responsible for the smaller surface pore size.  The authors have also described how the different process parameter such as foaming temperature, saturation pressure, and the membrane thickness has an effect on the morphology and mechanical properties of the membrane. The paper is written well. I want to accept the paper with minor revisions.

In the abstract, the authors have mentioned that the perforated membrane enhances the antifouling properties of the membrane. Do you have any data that suggests that the TPU perforated membrane enhances the antifouling?

In the material section, please provide the information about the type of TPU – the amount of the hard segment, amount of soft segment, type of polyol used for hard and soft segments, type of isocyanides used, and amount of hydrogen bonded polyurethane

Please provide the details about the homogeneous and the heterogeneous nucleation of CO2 in the polymeric matrix. Please add it in the introduction section. Please include how the enthalpy and entropic energy related to the activation energy of nucleation. Also, add how the activation energy reduces with the heterogeneous nucleation.

Please provide the reasons or some hypothesis about why does the higher amount CO2 mean higher nucleation sites. Please add it in section 3.2.

The line 307 says “ the melting enthalpy in these materials was about 40 J/g”, which material has melting enthalpy 40 J/gm PI or TPU? How it was measured?

Author Response

The manuscript entitled “Preparation of thermoplastic polyurethane (TPU) perforated membrane via CO2 foaming and its particle separation performance” describes the preparation of perforated membranes using CO2 as foaming agent. The methodology described is simpler than others described in the literature, however for its industrial application it would be necessary to go deeper into this methodology. The present work is well presented and structured. Therefore, in my opinion, the present manuscript can be recommended for publication after the author take into account the following remarks and perform the experimental work to establish their conclusions.

Comment 1: In Figure 2, the authors present the DSC results of the commercial polyurethane used. The temperature range is 25ºC and 225ºC, to observe the fusion of the hard segment of the material. However, the authors should include the DSC of the material at lower temperatures to observe the fusion of the soft segment. In this way we could know, in a certain way, the chemical composition of the soft segment and its influence on the formation of perforated membranes.

A1: As indicated in Figure 2, the melting peak of TPU generated by the crystallization of hard domains at 188.9 °C was about 1.24 J/g, which suggested that the crystallization of the used TPU was very week. In general, the soft domains of TPU could be crystallized, but its fusion of crystals was usually lower than that formed by the hard domains. Therefore, the formed crystals melting by soft domains would be very small, as a result, we did not find it using DSC thermograph actually. This paper mainly focused on the preparation of perforated TPU film by using an environmental friendly physical foaming technology, and the separation performance of the as-prepared material. In a separated research, we will investigate the sample preparation condition such as compression pressure during film preparation, the thermal behavior of polymer chain, and the foaming parameters such as saturation pressure, transfer time, foaming temperature, and foaming time on the cell density (both surface and fracture section), and to illustrate the cell nucleation and cell growth mechanism of TPU thin film during the foaming. This paper will be submitted very soon. 

Comment 2: The authors indicate that they introduce the polyurethane film into a hot PDMS bath to form the perforated membranes. However, the authors do not indicate if this process would lead to an adsorption of PDMS on the surface of the membrane. Is it necessary to remove siloxane residues? Please, the authors should comment on this aspect.

A2: We are sorry for the misleading. The PDMS bath with high temperature was used to foam TPU film. During the foaming, the two laminated PI films were still adhered on the surface TPU foam, and an additional force was required to separate the foam TPU film. Therefore, with the protection of PI film, the PDMS oil could be adsorbed by TPU film. These descriptions have been shown in the revised manuscript in lines 118-121 of page 3.

Comment 3: Throughout the manuscript, the authors have been carried out a study of cell sizes and their distribution. However, the authors do not mention how they have conducted these studies. Authors should indicate the used software and how they established the conditions for their studies and their interpretation.

A3: Thanks for your professional advice. This paper mainly focused on the preparation of perforated TPU film by using an environmental friendly physical foaming technology, and the separation performance of the as-prepared material. A separated study about the influences of polymer chains, preparation conditions, foaming parameters on the cell nucleation and growth mechanisms will be discussed.